The biomechanical characteristics of the strongman atlas stone lift

Hindle Benjamin 1 benjamin.hindle@student.bond.edu.au
http://orcid.org/0000-0001-8277-4664 Lorimer Anna 1 2
Winwood Paul 2 3
Brimm Daniel 4
http://orcid.org/0000-0001-9851-1068 Keogh Justin W.L. 1 2 5 6
1 Faculty of Health Sciences and Medicine, Bond University , Gold Coast, Queensland , Australia
2 Sports Performance Research Institute New Zealand (SPRINZ), Auckland University of Technology , Auckland , New Zealand
3 Faculty of Health, Education and Environment, Toi Ohomai Institute of Technology , Tauranga , New Zealand
4 Faculty of Medicine, University of Queensland , Herston, Queensland , Australia
5 Cluster for Health Improvement, Faculty of Science, Health, Education and Engineering, University of the Sunshine Coast , Sunshine Coast, Queensland , Australia
6 Kasturba Medical College, Mangalore, Manipal Academy of Higher Education , Manipal, Karnataka , India
Butcher Scotty
Electronic publication date: 2021 Sep 1
Publication date: 2021
Volume: 9
Electronic Location ID: e12066
Received 2021 Apr 8; Accepted 2021 Aug 5
Copyright: © 2021 Hindle et al.
Copyright year: 2021
Copyright holder: Hindle et al.
License: This is an open access article distributed under the terms of the Creative Commons Attribution License, which permits unrestricted use, distribution, reproduction and adaptation in any medium and for any purpose provided that it is properly attributed. For attribution, the original author(s), title, publication source (PeerJ) and either DOI or URL of the article must be cited.
License URL: https://creativecommons.org/licenses/by/4.0/

Keywords: Biomechanics, Strength-sports, Motion capture, Inertial devices, IMU, Weightlifting, Powerlifting, Strongman, Kinematics

Funding: Australian Government Research Training Program Scholarship This research was supported by an Australian Government Research Training Program Scholarship to the primary author. The funders had no role in study design, data collection and analysis, decision to publish, or preparation of the manuscript.

==============================
Background

The atlas stone lift is a popular strongman exercise where athletes are required to pick up a large, spherical, concrete stone and pass it over a bar or place it on to a ledge. The aim of this study was to use ecologically realistic training loads and set formats to (1) establish the preliminary biomechanical characteristics of athletes performing the atlas stone lift; (2) identify any biomechanical differences between male and female athletes performing the atlas stone lift; and (3) determine temporal and kinematic differences between repetitions of a set of atlas stones of incremental mass.

Methods

Kinematic measures of hip, knee and ankle joint angle, and temporal measures of phase and repetition duration were collected whilst 20 experienced strongman athletes (female: n = 8, male: n = 12) performed three sets of four stone lifts of incremental mass (up to 85% one repetition maximum) over a fixed-height bar.

Results

The atlas stone lift was categorised in to five phases: the recovery, initial grip, first pull, lap and second pull phase. The atlas stone lift could be biomechanically characterised by maximal hip and moderate knee flexion and ankle dorsiflexion at the beginning of the first pull; moderate hip and knee flexion and moderate ankle plantarflexion at the beginning of the lap phase; moderate hip and maximal knee flexion and ankle dorsiflexion at the beginning of the second pull phase; and maximal hip, knee extension and ankle plantarflexion at lift completion. When compared with male athletes, female athletes most notably exhibited: greater hip flexion at the beginning of the first pull, lap and second pull phase and at lift completion; and a shorter second pull phase duration. Independent of sex, first pull and lap phase hip and ankle range of motion (ROM) were generally smaller in repetition one than the final three repetitions, while phase and total repetition duration increased throughout the set. Two-way interactions between sex and repetition were identified. Male athletes displayed smaller hip ROM during the second pull phase of the first three repetitions when compared with the final repetition and smaller hip extension at lift completion during the first two repetitions when compared with the final two repetitions. Female athletes did not display these between-repetition differences.

Conclusions

Some of the between-sex biomechanical differences observed were suggested to be the result of between-sex anthropometric differences. Between-repetition differences observed may be attributed to the increase in stone mass and acute fatigue. The biomechanical characteristics of the atlas stone lift shared similarities with the previously researched Romanian deadlift and front squat. Strongman athletes, coaches and strength and conditioning coaches are recommended to take advantage of these similarities to achieve greater training adaptations and thus performance in the atlas stone lift and its similar movements.

Introduction

Strongman is a competitive strength-based sport where athletes perform heavier or more awkward/challenging variations of common activities of daily living or traditional tests of strength. Strongman exercises are often derived from traditional weight training exercises such as the clean and press, deadlift and squat (Harris et al., 2016). In a typical strongman competition event, an athlete may be required to lift large stones to various height ledges, carry weight-loaded frames, press large logs or dumbbells over-head or pull multi-ton vehicles such as trucks, buses or planes (Keogh & Winwood, 2017).

The atlas stone lift is a common strongman competition event which requires the athlete to pick up and place a large, spherical, concrete stone onto a ledge or over a bar (Fig. 1). The diameter of the stone, mass of the stone and height of the ledge/bar can vary between competitions and between competition classes which are typically based on sex and bodyweight. Common measures of performance in a competition atlas stone event is a maximum number of repetitions of a single mass stone over a bar in a timed period (usually 60 s); or the fastest time to place a series of stones (usually three to six stones) of incremental mass onto a ledge or over a bar.

Figure 1 An athlete performing the atlas stone lift.

Photo credit: Benjamin Hindle.

Qualitatively, the atlas stone lift has been suggested to share biomechanical similarity to various traditional weight training exercises (Hindle et al., 2019). The initial lift of the stone off the ground may be similar to lifting a sandbag or medicine ball off the ground using a Romanian deadlift technique; lifting the stone from the lapped position may be similar to the initiation of the concentric phase of a box squat from the seated position; and the final drive from a quarter-squat position to passing the stone over a bar/onto a ledge may be similar to the concentric phase of a barbell front squat where the load is positioned on the anterior surface of the body (Hindle et al., 2019)

Quantitative research into the biomechanics of athletes performing the atlas stone lift is limited, with the only study on this lift conducted to date analysing trunk muscle activation patterns and lumbar spine motion, load and stiffness (McGill, McDermott & Fenwick, 2009). Three experienced male strongman athletes (body mass: 117.3 ± 27.5 kg) performed a single lift of a 110 kg stone to a height of 1.07 m. When compared with other strongman lifts examined in the study, including the farmers walk, log lift, tire flip and yoke walk, the atlas stone lift was reported to result in the lowest lumbar spinal compression, which was suggested to be due to the athlete’s ability to curve their spine around the stone and keep the centre of mass of the stone close to their lower back (McGill, McDermott & Fenwick, 2009). The findings of McGill and colleagues were not, however, consistent with the retrospective injury study by Winwood et al. (2014b). In a survey of 213 male strongman athletes, the atlas stone lift was reported to account for the greatest percentage of injuries caused by common strongman exercises (including the yoke walk, farmers walk, log lift and tire flip) with the bicep and lower back being the most common sites of atlas stone lift injuries (Winwood et al., 2014b). The potential discrepancy in the findings of McGill, McDermott & Fenwick (2009) and Winwood et al. (2014b) may be due to the relatively light loads and low height to which the stone was lifted by athletes in the study by McGill, McDermott & Fenwick (2009), when compared with what would be lifted by athletes of similar body mass in training and competition today (load: >180 kg; height: 1 to >1.3 m).

Between-repetition comparisons of heavy, awkward lifting exercises performed in immediate succession (no rest period between repetitions), such as a series of atlas stone lifts are limited. Changes in biomechanics between repetitions have been observed due to an increase in load when performing the barbell back squat, whereby as load approaches an athlete’s one repetition maximum (1RM), greater trunk inclination and hip range of motion (ROM) has been observed (Yavuz & Erdag, 2017). The rest allocated between incremental load repetitions (loads of 80%, 90%, 100% 1RM; 5 min rest between each load) in Yavuz & Erdag (2017), should be noted as a distinct difference to a set of atlas stone lifts of incremental mass where minimal between-repetition rest periods typically occur during training and competition. Due to the differences in rest period and thus greater accumulation of acute fatigue in a series of atlas stone lifts when compared with squats performed in Yavuz & Erdag (2017), the transferability of the observations in Yavuz & Erdag (2017) to the atlas stone lift are still somewhat uncertain. Trafimow et al. (1993) demonstrated the effect of fatigue on the biomechanics of healthy male participants lifting loaded boxes (0–30 kg) from the floor to knuckle height. After performing an isometric half-squat hold (held until failure), participants employed more of a stoop lifting technique (straight leg) than a squat lifting technique (flexed knee), where the squat technique was preferentially used pre-fatigue. While qualitatively stoop and squat lifting techniques appear similar to components of the atlas stone lift, both the load (0–30 kg) and study population (healthy, recreationally active males) recruited in Trafimow et al. (1993) may make unclear whether such observations are transferable to the atlas stone lift performed by strongman athletes.

No studies have compared the biomechanics of male and female athletes performing the atlas stone or similar, heavy, awkward lifting exercises. A study by Lindbeck & Kjellberg (2001) observed between-sex differences in lower limb and trunk kinematics of office workers performing a stoop and squat lifting technique. Men exhibited greater trunk ROM for both lifting techniques, while female athletes exhibited greater knee ROM in the squat lifting technique (Lindbeck & Kjellberg, 2001). Similar to the box lifting study of Trafimow et al. (1993), the transferability of these observations to the atlas stone lift are uncertain due to the substantial difference in loading (male: 12.8 kg; female: 8.7 kg) and study populations (healthy office employees) compared to male and female strongman athletes performing the atlas stone lift. Of greater relevance to the atlas stone lift may be the studies of McKean & Burkett (2012) and Lisman et al. (2021), where between-sex kinematic differences were observed in trained persons performing the back squat (50% body mass) and over-head squat (un-loaded), respectively. In these studies, female athletes displayed a more upright trunk position during the overhead squat (Lisman et al., 2021) and back squat (McKean & Burkett, 2012) than male athletes. Male athletes displayed greater peak hip flexion in the overhead squat than female athletes (Lisman et al., 2021), while females displayed greater peak hip flexion in the back squat than male athletes (McKean & Burkett, 2012).

As this study is the first of its kind to estimate spatiotemporal and kinematic measures of male and female athletes performing the atlas stone lift, an emphasis is placed on the importance of undertaking a descriptive-type study of the movement pattern associated with the atlas stone lift. The aim of this study was to use ecologically realistic training loads and set formats to (1) establish the preliminary biomechanical characteristics of athletes performing the atlas stone lift; (2) identify any biomechanical differences between male and female athletes performing the atlas stone lift; and (3) determine temporal and kinematic differences between repetitions of a set of atlas stones of incremental mass. In alignment with the aim of the study it was hypothesised that: (1) various phases of the atlas stone lift will share biomechanical similarity with previously studied traditional weight training exercises; (2) differences in lower limb kinematics will be observed between male and female athletes, particularly at the hip joint; and (3) athlete biomechanics will change throughout the set, with greatest differences observed between the first and last repetition of the set.

By addressing this aim, researchers, strongman coaches and strength and conditioning coaches will be better equipped with the knowledge of the atlas stone lift biomechanics required to: provide strongman athletes with recommendation on how to perform the atlas stone lift based on the techniques of experienced strongman athletes; better prescribe strongman athletes with biomechanically similar exercises to the atlas stone lift for targeted training of specific phases of the lift; better prescribe the use of the atlas stone as a training tool for non-strongman athletes; and better structure future research into the strongman atlas stone lift.

Materials & methods

Experimental approach

A cross-sectional observational experimental design was used to describe the biomechanical characteristics of athletes performing the atlas stone lift and assess temporal and kinematic measures of an incremental mass, four atlas stone series. Well trained strongman athletes with strongman competition experience (Table 1) undertook two testing sessions. Session one consisted of a 1RM atlas stone lift to establish loading conditions for session two. Session two consisted of the collection of temporal and kinematic measures during three sets of four lifts of atlas stones of incremental mass (up to ~85% 1RM) over a fixed-height bar. Body mass, trochanterion-tibiale laterale height and tibiale laterale height anthropometric measures were taken by a trained person using ISAK methodologies (Marfell-Jones, Stewart & De Ridder, 2012) to assist in describing the study population.

Table 1 Participant characteristics.

Descriptor	Female	Male	
Age (years)	31.8 ± 6.5	31.8 ± 7.8	
Body mass (kg)	76.2 ± 15.4	115.6 ± 26.3	
Stature (m)	1.653 ± 0.43	1.811 ± 0.086	
Femur length (m)	0.399 ± 0.027	0.412 ± 0.045	
Tibia length (m)	0.470 ± 0.022	0.519 ± 0.031	
1RM atlas stone lift (kg)	80.3 ± 12.0	141.3 ± 24.9	
Strongman training experience (years)	2.1 ± 0.7	3.0 ± 1.7	
Strongman competition experience (number of competitions in past 2 years)	4.1 ± 2.8	3.5 ± 2.2	

Participants

Twenty experienced strongman competitors (12 male and eight female) were recruited from two local strongman gyms (Table 1). All participants were required to have a minimum of 18 months strongman training experience, have competed in a minimum of one strongman competition and be free from moderate or major injury for at least one week prior to testing. A moderate injury was defined as an injury that had stopped the athlete from performing a particular strongman exercise during a strongman session, while a major injury was defined as an injury which prevented the athlete from continuing with all exercises and/or the session completely (Winwood et al., 2014b; Keogh & Winwood, 2017). Participants meeting the above criteria were informed of the purpose of the study and asked to sign an informed consent form. Ethical approval was granted for all procedures used throughout this study by Bond University’s Human Research Ethics Committee (BH00045).

Trial conditions

To achieve optimal performance during the session, athletes were asked to prepare for each session in the same way in which they would prepare for a regular training session. Due to the range of individual loading parameters and experience level of all athletes recruited in the study, self-directed warm up routines were performed by each athlete (Winwood et al., 2014a, 2015a, 2015b; Renals et al., 2018; Winwood et al., 2019). Warm up routines lasted ~15–30 min and included repetitions of the atlas stone lift at loads approaching those expected to be used by the individual throughout the session. Generally, athletes would begin their warm up with dynamic stretching, including resistance band exercises, followed by barbell-only (no additional load) squats or deadlifts. Athletes would move on to stone pickups (either performing a Romanian deadlift-like pickup of the stone from the ground, or lifting the stone in a full range of motion to bar height without passing the stone over the bar) at low loading (~<60% 1RM). As athletes approached stone masses expected to be used in the session, the athlete would begin to complete full stone lift repetitions where the stone was passed over the bar. Athletes were permitted to use knee and elbow sleeves, lifting belts, arm/wrist wraps and tacky during sessions, as this is standard equipment used in competition and training.

Session protocols

Session one 1RM testing required athletes to lift a stone of greatest mass over a bar of fixed height (female: 1.2 m; male: 1.3 m). Athletes worked up to their heaviest stone in mass increments selected by the athlete. Mass increments were dependent on the mass of the stones available, the perceived effort of the previous lift and current training loads used by each participant. When an athlete failed to lift the stone over the prescribed height bar, the athlete was given one additional attempt to successfully complete the lift. Athletes were assigned rest periods of six to eight minutes between each stone attempt (Winwood, Keogh & Harris, 2011). The mass of the heaviest stone the athlete was able to successfully pass over the bar was determined to be their 1RM.

Session two was performed a minimum of seven days after session one and required athletes to perform three sets of a four stone series over a bar (female: 1.2 m; male: 1.3 m) as quickly as possible. Each stone within the series were of incremental mass, where stone one (repetition one) ≈60% 1RM, stone two (repetition two) ≈70% 1RM, stone three (repetition three) ≈80% 1RM and stone four (repetition four) ≈ 85% 1RM (Table 2). As is the nature of the atlas stone, stones were of a fixed mass (mass could not be added or removed from the stone), therefore stones within each series were selected based on the closest stone mass available to fit the required percentage of 1RM for each participant. The diameter and surface finish of stone varied with the mass of the stone (Table 2).

Table 2 Stone series characteristics.

Descriptor	Female	Male	
Stone one (repetition one)			
Mass (kg)	50.1 ± 7.3	90.7 ± 18.8	
% 1RM	62.6 ± 1.6	63.8 ± 4.3	
Diameter (m)	0.354 ± 0.015	0.428 ± 0.027	
Stone two (repetition two)			
Mass (kg)	55.8 ± 7.6	100.6 ± 20.0	
% 1RM	69.7 ± 2.0	70.9 ± 3.9	
Diameter (m)	0.369 ± 0.012	0.441 ± 0.034	
Stone three (repetition three)			
Mass (kg)	61.9 ± 8.5	110.7 ± 19.3	
% 1RM	77.3 ± 2.0	78.3 ± 4.3	
Diameter (m)	0.377 ± 0.020	0.455 ± 0.029	
Stone four (repetition four)			
Mass (kg)	69.0 ± 11.6	120.5 ± 21.9	
% 1RM	85.9 ± 3.0	85.2 ± 2.5	
Diameter (m)	0.394 ± 0.029	0.471 ± 0.036	

To begin each set, the athletes were positioned in the typical atlas stone competition starting position with the stone on the ground between their legs and their hands resting on the bar for which the stone was to be passed over. On the signal “athlete ready, three, two, one, lift” the participant commenced lifting stone one over the bar. After the completion of each repetition, the next stone in the series was positioned in front of the participant by a trained loading assistant. When an athlete was unable to pass a stone over the bar or the final stone in the series was successfully passed over the bar the trial was concluded, with each series typically completed in 60 s.

Data acquisition and analysis

Methodologies of Hindle, Keogh & Lorimer (2020) were used to estimate joint kinematics of athletes performing the atlas stone lift. Four magnetic, angular rate and gravity (MARG) devices (ImeasureU, Vicon Motion Systems Ltd., Oxford, UK) were used to capture acceleration, angular velocity (1,125 Hz) and magnetic field strength data (112 Hz). MARG devices were positioned on the pelvis (halfway between the left and right posterior superior iliac spine), right thigh (approximately 150 mm proximal to the lateral epicondyle of the femur), right shank (approximately 100 mm distal to the lateral tibial condyle) and right foot (midway between the base of the foot and the lateral malleoli) (Hindle, Keogh & Lorimer, 2020). The MARG data collected for each segment were input into a custom Matlab script (The Mathworks Inc., Natick, MA, USA) to measure hip, knee and ankle joint angles in the sagittal plane (Hindle, Keogh & Lorimer, 2020). The methodology has shown acceptable to excellent agreement with optical motion capture methodologies in similar movements such as the squat, box squat and sandbag pickup (Hindle, Keogh & Lorimer, 2020).

Two video cameras (iPad Air 2, iOS 13.3.1, Apple Inc., CA, USA) were used to capture video data at 30 Hz (Fig. 2). Video data were synchronised with MARG data using the ground impact of a submaximal jump performed immediately prior to the commencement of each set. The video data allowed for the calculation of the temporal parameters (phase duration, repetition duration), while joint kinematics at various instances throughout a repetition were obtained from the time-synched MARG data. Temporal and kinematic measurements assessed during each repetition of the atlas stone lift are defined in Table 3, with a pictorial representation of each phase of the lift presented in Fig. 3. Joint angle conventions are outlined in Fig. 4.

Figure 2 Schematic of equipment setup.

Figure 3 Atlas stone lift phase definition representation.

Figure 4 Joint angle definitions.

Table 3 Temporal and kinematic measurement definitions.

Parameter	Definition	
Recovery phase	Beginning: Stone set in front of the athlete (on ’lift’ call for first repetition in set or once stone is placed in front of the athlete and the loader is clear in subsequent repetitions)
End: Instance/final instance* of the athlete first touching the southern hemisphere of the stone	
Initial grip phase	Beginning: Instance/final instance* of the athlete first touching the southern hemisphere of the stone
End: Instance/final instance* of the stone leaving the ground	
First pull phase	Beginning: Instance/final instance* of the stone leaving the ground
End: Stone reaching peak positive trajectory prior to a negative trajectory toward the lap of the athlete.	
Lap phase	Beginning: Stone reaching peak positive trajectory prior to a negative trajectory toward the lap of the athlete.
End: Instance/final instance* of initial vertical movement of the stone from the lap position	
Second pull phase	Beginning: Instance/final instance* of initial vertical movement of the stone from the lap position.
End: > 50% of the stone passed over the bar.	
Joint angle	Hip, knee and ankle angle at the beginning and end of each phase. Joint angle definitions provided in Fig. 4. Positive angles denote flexion, negative angles denote extension.	
Hip ROM	Maximum angle between the pelvis and thigh minus minimum angle between the pelvis and thigh throughout a given phase.	
Knee ROM	Maximum angle between the thigh and shank minus minimum angle between the thigh and shank throughout a given phase.	
Ankle ROM	Maximum angle between the foot and shank minus minimum angle between the foot and shank throughout a given phase.	
Note:

*Final instance where multiple attempts were made to lift the stone off the ground.

Statistical methods

Data were checked for normality using visual inspection and a Shapiro Wilks test. Homogeneity of variances were checked using Levene’s test, homogeneity of covariances were checked using Box’s M-test (p < 0.001) and sphericity was checked throughout the computation of ANOVA tests. Mean and standard deviations of all variables were calculated for all phases throughout the stone lift. The joint kinematic results for the recovery and initial grip phases were not presented due to the high variability in the participants’ movements observed in these non-lifting, preparation phases, thus statistical analyses of these phases were not performed. A one-way repeated measures ANOVA test was used to establish the biomechanical characteristics of the lift by comparing: (1) between phase characteristics; (2) between repetition characteristics; and (3) between set characteristics. Between set statistical analysis was performed prior to further analyses to assess if data from each of the three sets could be combined. A two-way mixed model ANOVA test was used to identify interactions of sex and repetitions for each biomechanical characteristic. Partial eta-squared effect sizes (ηp2) were calculated for two-way interactions with classifications of negligible (ηp2 ≤ 0.01), small (0.01 > ηp2 ≥ 0.06), medium (0.06 > ηp2 ≥ 0.14) and large (ηp2 > 0.14) (Cohen, 1988). Bonferroni post-hoc pairwise t-tests were conducted on parameters where significant differences were detected. Cohen’s d (d) effect sizes were calculated for t-tests with classification of negligible (d < 0.2), small (0.2 ≤ d < 0.5), medium (0.5 ≤ d < 0.8) and large (d ≥ 0.8) (Cohen, 1988). Post-hoc intra-class correlation coefficient (ICC) and standard error of measurement (SEM) metrics were calculated to assess relative and absolute reliability of each biomechanical measure, respectively. Reliability was classified as poor (ICC < 0.5), moderate (0.5 ≤ ICC < 0.75), good (0.75 ≤ ICC < 0.9) and excellent (ICC ≥ 0.9) (Koo & Li, 2016). Statistical analyses were performed in R version 3.6.1 (R Development Core Team, Vienna, Austria), with statistical significance accepted at p < 0.05 unless otherwise stated.

Results

A total of 216, 236 and 232 repetitions were analysed for the hip, knee and ankle, respectively. The failure to analyse all joints throughout some repetitions was attributed to sensor malfunction (hip = 16; ankle = 4), sensor detachment (hip = 4) and two participants failing to complete all four stone repetitions within the set (stone/repetition four failed attempts: n = 4). Only full repetitions from successful lift off to lift completion were analysed.

General biomechanical characterisation—sex independent

The atlas stone lift could be characterised by: maximal hip and moderate knee flexion and ankle dorsiflexion at the beginning of the first pull and maximal hip ROM throughout the first pull; moderate hip and knee flexion and moderate ankle plantarflexion at the beginning of the lap phase and minimal hip, knee and ankle ROM throughout the lap phase; moderate hip and maximal knee flexion and ankle dorsiflexion at the beginning of the second pull phase and maximal knee and ankle ROM throughout the second pull phase; and maximal hip and knee extension and ankle plantarflexion at lift completion (Fig. 5, Tables S1, S2, S3).

Figure 5 Repetition independent joint kinematic and temporal measures.

(A) Hip joint kinematics; (B) knee joint kinematics; (C) ankle joint kinematics; (D) temporal measures of each phase.

Excluding the recovery and initial grip phases, the second pull phase was statistically longer in duration than all other lifting phases (0.27 ≤ d ≤ 1.12, p < 0.001), followed by the lap phase which was statistically longer in duration than the first pull phase (d = 0.34, p < 0.001) (Fig. 5, Table S3).

General biomechanical characterisation—sex dependent

When compared with male athletes, female athletes exhibited: greater hip flexion and ankle plantarflexion at the beginning of the first pull (0.78 ≤ d ≤ 1.21, p < 0.001) and greater overall hip ROM throughout the first pull (d = 0.56, p < 0.001); greater hip flexion and knee extension at the beginning of the lap phase (0.58 ≤ d ≤ 0.77, p < 0.001), and smaller hip and ankle ROM throughout the lap phase (0.26 ≤ d ≤ 0.46, p ≤ 0.049); greater hip flexion, knee extension and ankle plantarflexion at the beginning of the second pull phase (0.29 ≤ d ≤ 0.48, p ≤ 0.034), and smaller knee ROM and greater ankle ROM throughout the second pull phase (-0.53 ≤ d ≤ 0.32, p ≤ 0.021); and greater hip flexion and ankle plantarflexion at lift completion (0.41 ≤ d ≤ 0.85, p ≤ 0.003) (Fig. 5, Tables S1, S4).

Few statistical between-sex temporal differences were observed (Table S5). Male athletes displayed a statistically longer second pull phase duration than female athletes (d = 0.42, p = 0.012) (Fig. 5, Tables S1, S4).

Between repetition biomechanical differences—sex independent (main effect)

Statistically significant between-repetition differences were most commonly observed for joint kinematics between combinations of the first two repetitions and the last two repetitions of the set (e.g., between repetition one-two and three-four) (Figs. 6, 7, 8, 9, Table S5). First pull phase hip and ankle ROM was smaller in repetition one than the final three repetitions (−0.717 ≤ d ≤ −0.496, p ≤ 0.002) (excluding repetition two ankle ROM). Lap phase hip and ankle ROM was smaller in repetition one than the final three repetitions (−1.15 ≤ d ≤ −0.46, p < 0.001), and smaller in repetitions two and three (hip only) than repetition four (−0.65 ≤ d ≤ −0.37, p ≤ 0.003). No statistical between-repetition differences were observed at any joint for the position in which athletes began the second pull phase (Table S5).

Figure 6 Sex and repetition dependent joint ROM kinematic measures for each phase, (A–C) hip joint kinematics; (D–F) knee joint kinematics; (G–I) ankle joint kinematics.

Figure 7 Sex and repetition dependent hip joint kinematic measures for beginning/end of each phase.

(A) Hi beginning first pull; (B) Hip beginning lap; (C) Hip beginning second pull; (D) Hip lift completion.

Figure 8 Sex and repetition dependent knee joint kinematic measures for beginning/end of each phase.

(A) Knee beginning first pull; (B) Knee beginning lap; (C) Knee beginning second pull; (D) Knee lift completion.

Figure 9 Sex and repetition dependent ankle joint kinematic measures for beginning/end of each phase.

(A) Ankle beginning first pull; (B) Ankle beginning lap; (C) Ankle beginning second pull; (D) Ankle lift completion.

For each repetition, individual phase durations and total repetition duration increased as the set progressed (Fig. 10, Table S6), with medium to large effect sizes recorded between repetition one and repetitions three and four (0.64 ≤ d ≤ 1.73, p ≤ 0.003). Where statistical differences were reported for phase duration between sequential stones (e.g., repetition one vs repetition two, repetition three vs repetition four), smaller effect sizes were typically observed (0.31 ≤ d ≤ 1.03, p ≤ 0.005) (Table S6).

Figure 10 Sex and repetition dependent temporal measures.

(A) Recovery phase; (B) Initial grip phase; (C) First pull phase; (D) Lap phase; (E) Second pull phase; (F) Entire repetition.

Between repetition biomechanical differences—sex dependent (two-way interaction)

While not evident in female athletes, male athletes generally displayed: smaller hip ROM during the second pull phase of the first three repetitions when compared with the final repetition (−0.87 ≤ d ≤ −0.59, p ≤ 0.011); smaller hip extension at lift completion during the first two repetitions of the set when compared with the final two repetitions (−1.24 ≤ d ≤ −0.55, p < 0.038); and greater plantarflexion of the ankle at lift completion in the first repetition when compared with the final repetition (d = 0.75, p = 0.014) (Tables S5, S6, S1). No temporal two-way interactions between sex and repetition were observed (Table S5).

Between set biomechanical differences

Between-set analysis was performed so to identifying any potential effects of set number on the biomechanics of the athlete. Hip flexion was greater at the beginning of the first pull, lap phase and second pull in set one than set two and three (0.04 ≤ d ≤ 0.26, p ≤ 0.013) (Tables S7, S8, S9). Second pull duration was significantly greater during set one than set three (d = 0.19, p = 0.012) (Tables S8, S9). No statistical between-set difference in total repetition duration was observed for any repetition.

Discussion

In alignment with the descriptive nature of the research, the aim of this study was to use ecologically realistic training loads and set formats to (1) establish the preliminary biomechanical characteristics of athletes performing the atlas stone lift; (2) identify any biomechanical differences between male and female athletes performing the atlas stone lift; and (3) determine temporal and kinematic differences between repetitions of a set of atlas stones of incremental mass.

General biomechanical characterisation—sex independent

To describe the general movement pattern of the atlas stone lift, hypothesis one sought to determine if the various phases of the atlas stone lift were biomechanically similar to selected traditional weight training exercises.

Recovery and initial grip phase

Only temporal parameters were measured for the recovery and initial grip phase due to the high variability in joint kinematics observed during data collection and upon review of video data. This variability included athletes repositioning the stone by foot, and various individual set-up routines. The recovery and initial grip phases may be viewed as ‘preparation’ phases where the stone is yet to be physically lifted from the ground. These phases may be analogous to the athlete approaching the bar and first touching the bar in a 1RM deadlift, or the phase which may be defined between when an athlete returns the bar to the ground before lifting it back up in an as many repetitions as possible (AMRAP) deadlift event.

First pull phase

The beginning of the first pull phase of the atlas stone lift was characterised by maximal hip flexion and moderate knee flexion and ankle dorsiflexion. The maximal hip flexion (72.7 ± 20.0°) at the beginning of the first pull phase was similar to that of the maximal hip flexion occurring during the Romanian deadlift (79.97 ± 15.85°) (Lee et al., 2018). Knee flexion at the beginning of the first pull in the atlas stone lift (45.6 ± 12.7°) was however, slightly larger than the knee flexion reported for the Romanian deadlift (33.86 ± 12.59°) (Lee et al., 2018). The relative similarity in the starting position of the atlas stone lift to the Romanian deadlift in conjunction with previous research on the trunk muscle activation patterns of athletes performing the atlas stone lift (McGill, McDermott & Fenwick, 2009) and the Romanian deadlift (Delgado et al., 2019), suggest that performing the first pull phase of the atlas stone lift may result in similar training adaptations to the Romanian deadlift. Schellenberg et al. (2013) reported similar maximal hip flexion (75.3 ± 9.2°) when athletes performed goodmornings with an external barbell load of 25% body mass. Where an athlete is required to focus on strengthening the hamstrings or is unable to perform either the atlas stone lift or Romanian deadlift due to specific injuries which prevent grasping a stone or barbell, goodmornings may be a suitable accessory exercise.

The first pull phase of the atlas stone lift was statistically shorter in duration than all other lifting phases (1.043 ± 0.360 s) and involved the largest ROM of the hip and second largest knee ROM of all phases. This indicates that a rapid extension of the hip and knee is key in initiating movement of the stone from the ground to a position close to the athlete’s chest and centre of mass (COM) at the beginning of the lap phase. Training for power and rate of force development during rapid extension of the hip and knee and to a lesser extent the ankle (in exercises such as the power clean or other weightlifting derivatives) may promote the physiological adaptations required for greater performance throughout the first pull phase of the atlas stone lift (Winwood, Keogh & Harris, 2011; James et al., 2020).

Lap phase

At the beginning of the lap phase, the athlete is generally in a position of moderate hip (24.0 ± 18.1°) and knee flexion (45.1 ± 17.6°), and moderate ankle plantarflexion (−3.7 ± 8.5°), supporting the lower portion of the stone with the hands and arms. For the majority of the athletes, gripping the stone with the hands on the lower portion of the stone throughout the entirety of the lift provided insufficient clearance to pass the stone over the bar upon standing with full extension of the hips and knees and an anatomical ankle position. To overcome this, athletes typically attempted to pull the stone as high as possible toward the chest at the end of the first pull/start of the lap phase, before retrieving and resting the stone in the lap. Whilst in the lap, the athlete re-gripped the stone with the arms and hands hugging the upper portion of the stone. The relatively large variance in the duration of the lap phase (1.325 ± 1.112 s) was representative of the time some athletes invest in ensuring a secure grip of the stone, whereby failing to grip the stone may result in dropping the stone during the second pull phase, costing the athlete time and energy in re-attempting the lift.

Two athletes used a “zero-lap” phase technique (commonly referred to as a “one-motion” technique within the strongman community) for the first two repetitions of each set, whereby the stone was lifted in a single motion with no transition of grip, no negative trajectory of the stone and thus, no lap phase. Employing the zero-lap technique likely reduces the total duration of the repetition. The two athletes that used this technique were the tallest athletes, indicating a possible advantage for taller athletes when lifting stones of lower mass (relative to 1RM) to/over an object of the same absolute height.

A short ROM, double knee bend technique was used sporadically by some athletes to initiate a stretch shortening cycle just prior to the beginning of the second pull phase. While the stretch-shortening cycle is commonly used in weightlifting events to ensure maximal force and power can be rapidly applied to the barbell (Enoka, 1979; Gourgoulis et al., 2000; Winwood et al., 2015b), evidence supporting its effectiveness for heavy/strength-based lifts performed over an extended duration, such as the atlas stone lift, is conflicting (McBride et al., 2010; Swinton et al., 2012).

Second pull phase

Moderate hip (40.2 ± 22.5°) and maximal knee (70.0 ± 20.7°) flexion and ankle dorsiflexion (10.3 ± 10.3°) at the beginning of the second pull phase and maximal knee (65.2 ± 20.1) and ankle (35.0 ± 12.7°) ROM throughout the second pull phase were observed for the atlas stone lift.

The concentric movement of the stone throughout the second pull phase, with the load positioned in front, has been qualitatively suggested to share kinematic characteristics with the front squat (Hindle et al., 2019). The front squat has, however, been characterised by greater hip (94.2 ± 22.4°) and knee (125.1 ± 12.6°) flexion at the beginning of the concentric phase than the atlas stone lift (Krzyszkowski & Kipp, 2020). Where greater strength adaptations may be achieved by performing an exercise with increased ROM (Bloomquist et al., 2013), strongman coaches may consider using the front squat in the training programs of strongman athletes to target the general knee and hip extension requirements of the atlas stone lift through a greater ROM, thus encouraging greater strength adaptations.

The final instance of the second pull phase (lift completion) demonstrates the triple extension of the hip and knee and plantarflexion of the ankle to a position where the athlete is in an almost-neutral standing position (hip: 6.1 ± 14.0°; knee: 8.4 ± 10.0°; ankle: −10.7 ± 18.1°). Although only quantifiable in the current study by the variance in kinematic measures, this rapid triple extension appeared to visually vary within and between athletes. For example, some athletes were able to perform the triple extension with enough power and timing to project or ‘pop’ the stone off their chest and onto/over the bar. In the pop technique, the athlete qualitatively appeared to lift the stone at a normal rate from the beginning of the second pull phase, before quickly extending the hip and spine toward the end of the second pull phase. As a result of the rapid movement of the stone towards the end of the second pull phase, the stone appears to ‘pop’ off the athlete’s chest and pass over the bar without the athlete remaining in contact with the stone. On the other hand, athletes who had to ‘grind’ the stone over the bar, displayed a substantial decrease in vertical stone velocity as the centre of mass of the stone approached the height of the bar. These athletes sometimes exhibited both hip extension and ankle plantarflexion as the stone passed over the bar. Athletes using the grind technique appeared to have to apply a force to the stone up until the precise moment at which the stone passed over the bar. In alignment with hypothesis one, some biomechanical similarity was present between phases of the atlas stone lift and traditional weight training exercises including the Romanian deadlift and front squat.

General biomechanical characterisation—sex dependent

A number of between-sex differences in joint kinematics were observed. Most notably, female athletes exhibited greater hip flexion (female: 84.7 ± 18.7°; male: 63.7 ± 15.8°) and ankle plantarflexion (female: 0.3 ± 8.4°; male: 6.0 ± 6.0°) at the beginning of the first pull, lap and second pull phase than male athletes.

The between-sex difference in hip flexion at the beginning of the first pull may be the result of the differences in anthropometric ratios of the female and male population. At the beginning of the first pull, a greater arm to lower limb length ratio would enable an athlete to grip the bottom of the stone with less flexion of the hip (assuming constant knee flexion angle). Keogh et al. (2008) reported statistically greater arm to leg length ratios in male powerlifters (67.8% ± 2.9%, n = 54) when compared with female powerlifters (64.5% ± 2.5%, n = 14), supporting the deduction that the between-sex differences observed in hip flexion at lift off for the atlas stone lift may be partially due to the anthropometric differences between male and female strength athletes.

The smaller hip flexion displayed by male athletes at the beginning of the lap and second pull phase may be a mechanism used by male athletes to accommodate the larger diameter stone (typically lifted by male athletes when compared with female athletes) so to ensure the COM of the stone remains as close as possible to their COM and within their base of support. The compensative mechanism of greater hip extension may result in a similar stone to body COM distance and thus resistive moment arm length about the lumbar spine in male and female athletes. Although not measurable in the current study, such a result has been reported in a study in which males had significantly greater absolute but not relative L5/S1 joint moments than females when lifting boxes between 15–24 kg from a pallet at a self-selected pace (Plamondon et al., 2014). The between-sex differences in hip, knee and ankle joint kinematics and phase duration measures observed while athletes performed the atlas stone lift are in support of hypothesis two.

Between repetition biomechanical differences—sex independent (main effect)

Hip and ankle joint ROM during the initial pull and lap phase of the lift were generally smaller for athletes during repetition one when compared with the final three repetitions. Greater flexion of the knee and hip at the beginning of the first pull were generally observed in the first two repetitions when compared with the final two repetitions.

The smaller hip and ankle ROM in the initial repetitions than the later repetitions indicate athletes performed abbreviated versions of the lift to begin the set. The strategy of athletes performing an abbreviated version of the lift is likely executed with the intention of self-preservation of energy (Hooper et al., 2014) and conservation of overall repetition and set time. This is supported by the statistically shorter phase durations and total repetition duration observed during the first two repetitions when compared with the final two repetitions of the set. The increased hip ROM when lifting the greater mass stones is also in line with previous research on load-dependant biomechanical differences observed during the back squat (Yavuz & Erdag, 2017).

Although fatigue was not directly measured in this research, the very short recovery duration between each repetition may contribute to some level of athlete fatigue. Recovery phase duration was found to increase as athletes progressed through the set of four atlas stone lift repetitions. Where the onset of fatigue is observed, research has demonstrated significant changes in joint kinematics of male participants performing a box lifting task (Trafimow et al., 1993). Such previous research may suggest that some of the between repetition differences observed in the current study be due to the acute effect of fatigue that progressively increased within the set of incremental mass stone lifts. In support of hypothesis three, a number of between-repetition differences were observed in athletes performing the atlas stone lift. Further, a large portion of between-repetition differences observed were between repetition one and four.

Between repetition biomechanical differences—sex dependent (two-way interaction)

Male athletes exhibited smaller hip ROM during the second pull phase of the first three repetitions when compared with the final repetition and smaller hip extension at lift completion during the first two repetitions of the set when compared with the final two repetitions. Female athletes appeared to use a more consistent technique throughout the four repetitions, whereby they did not exhibit these significant between repetition differences.

To ensure the bottom of the stone cleared the height of the bar in the final two repetitions, male athletes appeared to use greater extension (often hyperextension) of the hip. The greater extension of the hip at lift completion, likely contributed to the greater hip ROM displayed by male athletes in the final repetition when compared to the first three repetitions.

While the two-way interactions between sex and repetition further support hypothesis three, the exact reasoning behind the different mechanisms used throughout the set by male and females is somewhat unclear. Future researchers may look to investigate how between-sex differences in anthropometry, motor control and muscle recruitment strategies contribute to the kinematic between-sex differences observed during the atlas stone lift series.

Additional considerations

The current study is not exempt from limitations. As with any research, care should be taken when interpreting comparative results between groups, ensuring the magnitude of the error of the measurement system is recognised. In the case of the temporal parameters, the measurement accuracy was limited by the frame rate of the video camera, while kinematic parameters were limited by the accuracy of the MARG-based motion capture methodology (Hindle, Keogh & Lorimer, 2020). Good (ICC ≥ 0.75) to excellent (ICC ≥ 0.9) relative reliability was generally found for all biomechanical parameters measured within the study using the MARG and video camera methods (Table S10).

Twenty experienced strongman athletes (12 male, eight female) were recruited for the study. While the combined number of male and female strongman athletes recruited in the current study is much larger than the number of strongman athletes recruited in any previous strongman exercise biomechanics study, the individual number of male (n = 12) and female (n = 8) participants is similar or only slightly larger than previous research (McGill, McDermott & Fenwick, 2009; Keogh et al., 2010a; Keogh et al., 2010b; Keogh et al., 2014; Winwood et al., 2014a, 2015a, 2015b; Renals et al., 2018). A greater number of both male and female athletes would strengthen the conclusions drawn from the observed between-sex biomechanical differences.

Variation in the increments of the mass of the stones, dimensions of stones and surface finish of stones may also be viewed as a limitation to this study. Variable increments, dimensions and surfaces of stones, is however a reality of the sport of strongman and provides greater insight into the realities of strongman biomechanics.

As this is the first biomechanics study to describe kinematic and temporal parameters of athletes performing the atlas stone lift there is much scope for future research, including: transverse and frontal plane joint kinematic analyses; establishing relationships between anthropometrics of strongman athletes and their biomechanical characteristics; the effect of stone dimension, mass and surface finish on the biomechanics of an athlete; the injury risks associated with the atlas stone lift; and the biomechanical determinants of greater performance in the atlas stone competition event.

Conclusions

This study provides the first kinematic and temporal description of male and female athletes performing the atlas stone lift using set and repetition schemes that are commonly used in strongman training. The atlas stone lift could be biomechanically characterised by a recovery, initial grip, first pull, lap and second pull phase. Between-sex biomechanical differences were suggested to be, in-part, due to anthropometric differences between sexes, while between-repetition differences may be attributed to increases in stone mass as well as some acute fatigue that increased throughout the set. Strongman athletes, coaches and strength and conditioning coaches are recommended to take advantage of the similarity shared between the atlas stone lift and the identified traditional weight training exercises so to achieve greater training adaptations and thus performance in the atlas stone lift and its similar traditional weight training movements.

Supplemental Information

Supplemental Information 1 Supplementary Tables.

Click here for additional data file.

The Authors wish to acknowledge Jean-Stephen Coraboeuf, Colin Webb and Greg Nuckols for their input in designing testing protocols; Evelyne Rathbone for her assistance in advising on the statistical methods used; Robert Palmer and Simon Hiroyuki Welch for their assistance in collecting data; and Coco’s Gym Gold Coast and Panthers Powerlifting Gym Brisbane for supplying equipment and accommodating data collection sessions.

Additional Information and Declarations

Competing Interests

Author Contributions

Human Ethics

Data Availability

Justin Keogh is an Academic Editor for PeerJ. The authors declare no other competing interests associated with the publication of this article.

Benjamin Hindle conceived and designed the experiments, performed the experiments, analyzed the data, prepared figures and/or tables, authored or reviewed drafts of the paper, and approved the final draft.

Anna Lorimer conceived and designed the experiments, analyzed the data, authored or reviewed drafts of the paper, and approved the final draft.

Paul Winwood conceived and designed the experiments, analyzed the data, authored or reviewed drafts of the paper, and approved the final draft.

Daniel Brimm conceived and designed the experiments, performed the experiments, analyzed the data, authored or reviewed drafts of the paper, and approved the final draft.

Justin W. L. Keogh conceived and designed the experiments, analyzed the data, authored or reviewed drafts of the paper, and approved the final draft.

The following information was supplied relating to ethical approvals (i.e., approving body and any reference numbers):

Bond University Human Research Ethics Committee approved this research (BH00045).

The following information was supplied regarding data availability:

The raw data is available at Harvard Dataverse: Hindle, Benjamin, 2021, “Strongman biomechanics—atlas stone IMU data”, https://doi.org/10.7910/DVN/PQOIGN, Harvard Dataverse, V1, UNF:6:XeAdgDKTvpRWFbedFLDmWg== [fileUNF].

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
