# Peer review of "The biomechanical characteristics of the strongman atlas stone lift"

_PeerJ, doi:10.7717/peerj.12066_

## Round 0.1 · original submission · Minor Revisions

Thank you for your patience as I sorted out reviewers. Both reviewers have suggested very minor edits to your paper, which is very well written and will contribute greatly to the literature in this area. Please attend to each of the reviewers' comments in detail upon resubmission.
Cheers,
Scotty

·

Basic reporting

I believe this study meets all the standards of Peer J reporting.

Experimental design

The experimental design meets all the standards of Peer J.

Validity of the findings

I have a few comments on the discussion. I put my notes to the author in that section. I do believe the validity of the findings meets all the standards of Peer J.

Additional comments

Great study on a topic that does not have much literature on it. The authors did a nice job on recruiting a trained sample and used validated methods for the kinematic analysis. I see the statistics were professionally done as well.

This paper is well written and is easy to understand. I do not see any methodological flaws in the design or the data analysis. I feel the discussion linking the exercises to the joint positions does a nice job on creating a bridge between which exercises someone should do to improve the phases of performance for the atlas stone lift. Lastly, I agree with the analysis that different arm to lower body length ratios are why females had a greater degree of hip flexion at the second pull phase

I have two questions about a reference in the Introduction and a measurement in the Discussion. In addition, I have a small suggestion for the authors in the Discussion.


Minor Change: Introduction, Line 138: Does the author mean to reference Lisman et al (2021) as opposed to Lindbeck and Kjellberg, 2001)? Because in the previous sentence, they refer to the females have a more upright position in the OHS, which is what Lisman et al (2021) examined. I don’t have the paper in front of me but based on the abstract, I don’t think Lindbeck examined OHS.



Minor change: Discussion, Line 389-409: This is just a suggestion. For the first pull phase, the author makes a connection between the RDL as an exercise similar to the atlas stone lift due to similar hip position. Yes, I agree. My additional suggestion is the good morning exercise. The hip position is similar (70-750) and activates comparable muscles. I understand the loading is on the back for the GM and not below hips therefore, the RDL is much better. However, this can be a secondary option if the athlete can’t perform RDL due to injury. For example, if the athlete had a hand injury. I have put a reference below on GM hip position.

Schellenberg et al. Kinetic and kinematic differences between deadlifts and goodmornings. BMC Sports Science, Medicine, and Rehabilitation 2013, 5:27 http://biomedcentral.com/2052-1847/5/27

Minor change: Discussion, Line 475-480: I agree with the author that the reason for greater hip flexion in the females at the beginning of the first pull phase is likely due to a greater arm-leg length ratio for males compared to females. My question is, why didn’t they measure this? They measured tibia length and femur length in Table 1, it seems to make sense to measure arm length as well, therefore you can see if this actually is the case. As opposed to deducing from Keogh (2008) study.

Reviewer 2 ·

Basic reporting

Line 204- When describing the heaviest stone lifted, consider using different terminology than “maximum mass stone”
Line 453- Replace “sees” with either “demonstrates” or “represents”.
Line 456- Replace the word “powerful” with a temporal descriptor since “power” was not measured in this study.
Line 458- Qualitatively explain “pop”
Line 460- Qualitatively explain “grind”..
The authors of this studies provided the requisite amount of references to the background and context provided.
Lines 488-489- Consider citing a study which performed a similar task (e.g., deadlift or stoop lifting) and provided between sex-differences in lumbar spine net joint moments).

This article was structured well, all tables and figures were provided. Raw data was provided in the supplemental materials.

The relevant results of this study were consistent with the stated hypotheses.

Experimental design

This study was the first of its kind to explore the biomechanical characteristics of the Atlas stone lift and adhered to the Aims and Scope of the journal.

The research question was well defined, relevant and meaningful for individuals interested in learning the biomechanical characteristics and requisites of the Atlas stone lift.

This study provided all data collection methods and relevant information to the equipment used (e.g., sampling frequencies, equipment names, and model numbers). The authors also provided citations for the data collection methods demonstrating the validity of the methods and use in previous research.

For the most part, the methods were described with sufficient detail and information to replicate. However, the warmup protocols (lines 196-196) were ambiguous and not controlled. A brief general description of the warmups would allow future researchers to better replicate the data collection procedures.

Validity of the findings

The rationale and benefit to the literature was clearly stated. The authors provided future researchers with a method of assessing atlas stone lifting biomechanics and discussed the limitations of this study which provide future researchers with an opportunity to further contribute to the literature pertaining to this topic of the study.
All underlying data have been provided are robust and statistically sound. Due to the data analysis issues related to the variability in certain phases of the Atlas Stone lift, Absolute and Relative reliability (intra-class correlation (ICC) and coefficient of variations (CV) or standard error of measurement (SEM) should be provided to ensure that the phases analysis demonstrated sufficient reliability.
Conclusions were well stated and were link to the original research question and limited to the supported results.

Additional comments

Dear authors, thank you for providing me with the opportunity to review your manuscript which sought to explore Atlas stone lift kinematics and kinematic differences between males and females. The introduction provided a concise yet informative explanation about strongman biomechanics, the atlas stone lift, associations with similar resistance training exercises and the gap in the literature.

---

## Round 0.2 · accepted · Accept

Dear authors, thank you for your attention to detail in submitting your revision and rebuttal. Incidentally, I greatly appreciate your response style (ie. green shading and reference to the change in the manuscript). Congratulations on a great study! Scotty